# A Review of How Uncertainties in Management Decisions Are Addressed in Coastal Louisiana Restoration

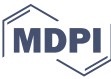

**Angelina M. Freeman \*, James W. Pahl**  **, Eric D. White**  **, Summer Langlois, David C. Lindquist, Richard C. Raynie and Leigh Anne Sharp**

Louisiana Coastal Protection and Restoration Authority, Baton Rouge, LA 70802, USA;
James.Pahl@LA.GOV (J.W.P.); Eric.White@la.gov (E.D.W.); Summer.Langlois@la.gov (S.L.);
David.Lindquist@la.gov (D.C.L.); Richard.Raynie@la.gov (R.C.R.); LeighAnne.Sharp@LA.GOV (L.A.S.)
\* Correspondence: Angelina.Freeman@la.gov; Tel.: +1-225-342-3790

**Abstract:** Louisiana has lost over 4800 km$^2$ of coastal land since 1932, and a large-scale effort to restore coastal Louisiana is underway, guided by *Louisiana's Comprehensive Master Plan for a Sustainable Coast*. This paper reviews science-based planning processes to address uncertainties in management decisions, and determine the most effective combination of restoration and flood risk reduction projects to reduce land loss, maintain and restore coastal environments, and sustain communities. The large-scale effort to restore coastal Louisiana is made more challenging by uncertainties in sediment in the Mississippi River, rising sea levels, subsidence, storms, oil and gas activities, flood-control levees, and navigation infrastructure. To inform decision making, CPRA uses structured approaches to incorporate science at all stages of restoration project planning and implementation to: (1) identify alternative management actions, (2) select the management action based on the best available science, and (3) assess performance of the implemented management decisions. Applied science and synthesis initiatives are critical for solving scientific and technical uncertainties in the successive stages of program and project management, from planning, implementation, operations, to monitoring and assessment. The processes developed and lessons learned from planning and implementing restoration in coastal Louisiana are relevant to other vulnerable coastal regions around the globe.

**Keywords:** ecosystem restoration; coastal and adaptation planning; estuaries; Mississippi River Delta





## 1. Introduction

River deltas are some of the most ecologically and economically productive and highly populated environments in the world, and also some of the most vulnerable [1,2]. Low-lying landscapes built at river mouths, the architecture of a delta is the product of both a river system and its sediments, as well as the physical processes of the receiving basin [3]. The world's deltas formed during the last 6500–8500 years as sea level rise stabilized after the last glacial maximum ~18,000 years ago [4,5]. Changes in environmental conditions both in the drainage basin and the receiving basin can profoundly transform the delta plain, and lead to advancement or retreat of the delta, aggradation, or delta switching. Subsidence from both natural compaction processes and anthropogenic factors, such as fluid withdrawal, increases the relative sea level rise of these coastal landscapes. Reductions in sediment supply due to upstream levees and dams lessen mitigation of subsidence [6], with an estimated 25 million people living on sediment-starved deltas [7]. Although 11,000 deltas globally have experienced net land gain over the past 30 years from increases in river fluvial sediment content from deforestation, as well as redistribution of sediments on deltas transitioning toward increasingly tide-dominated, these gains are not expected to be sustained under rising sea level conditions [8–11] and projected declines in fluvial sediment delivery [12]. Many of the world's deltas are also population centers, with the majority of megacities located on deltas [13]. These deltas are impacted by human

habitation and urbanization, and by increased subsidence from large-scale engineering projects and resource extraction [14]. Flooding of these highly populated delta surface areas is estimated to increase by 50% in the twenty-first century [9].

Coastal Louisiana, largely built by the Mississippi and Atchafalaya Rivers over the last ~7000–8000 years, lost an estimated 4833 km$^2$ of its land area from the early 1930s to 2016 [15]. The Mississippi River Delta is one of the most ecologically important habitats in North America [16], providing habitat for millions of migratory waterfowl [17] and supporting highly productive fisheries [18] and tourism. The top port (by tonnage) in the United States is the Port of South Louisiana, positioned in the Mississippi River Delta [19], and Louisiana is a major supplier of oil and natural gas [20]. A diverse landscape of freshwater, brackish, and saline wetlands, barrier islands, coastal bays, low-relief uplands, and ridges, coastal Louisiana is home to over two million people, including unique coastal cultures. Projections of future coastal Louisiana land loss, in the absence of sediment input, range from 10,000 km$^2$ to 13,500 km$^2$ by 2100 [21] to 3126 km$^2$ to 10,679 km$^2$ in 50 years [22]. Building on the current trajectories, undertaking no additional restoration action is projected to lead to both ecological and socio-economic catastrophe in coastal Louisiana's deltaic plain.

Ecosystem sustainability is often constrained in highly-engineered deltaic systems worldwide [23]. Anthropogenic interventions such as levees, dams, and channel deepening for navigation control the development of many deltas currently, moving the morphology away from a natural state [10], and increasing the importance of understanding delta instabilities such as bank failure and avulsion [24,25]. Restoration of deltaic systems can be complicated by sediment and nutrient loads that are often affected by actions thousands of kilometers upstream [26]. Although the scientific knowledge of deltaic systems is advanced, knowledge about restoration processes in these complex systems is comparatively nascent. Integrating science into environmental decision making can be difficult, and the time between detecting a problem and political action can be lengthy [27–30]. This presents additional challenges as environmental changes are occurring at accelerated rates due to human impacts. There is a growing need for adaptation planning, and for chronicling and evaluating planning processes, which are not well known among practitioners and the scientific community [31].

The transdisciplinary application of science and engineering necessary to restore and protect coastal Louisiana is pioneering. This paper reviews, describes, and evaluates the processes Louisiana's Coastal Protection and Restoration Authority (CPRA) has developed and uses to make science-based management decisions in uncertain conditions, with input from stakeholders, to restore a low-lying coastal ecosystem. This information has previously mostly only been available in governmental reports, which were reviewed and synthesized in combination with institutional knowledge for this paper. Successful management requires integrating science into decision making to support the strategic implementation of restoration programs and projects. Addressing and reducing uncertainties in management decisions and finding solutions to ecosystem restoration and human community resilience that are sustainable in the face of continued subsidence, rising sea levels and repeated storm impacts driven by human activities, including climate change, are critical to the continued environmental and economic productivity of Louisiana and the Gulf of Mexico region, and other vulnerable coastal deltaic ecosystems worldwide.

## 2. Scientific Framework and Management Needs

### 2.1. Scientific Framework of the Mississippi River Delta

The Mississippi River drains approximately 41% of the United States, and is the largest river system in North America, transporting sediment from the continental interior of its 3.4 million km$^2$ drainage basin to the delta and coastal ocean [32]. The seventh largest river in the world by sediment load, the Mississippi River has supplied vast amounts of sediment to the northern Gulf of Mexico, and built the Mississippi River Delta since sea level stabilized ~7000 years ago [33]. Historically a meandering type of river system, the

Mississippi River has seasonal flood and non-flood discharge cycles, with mean spring discharges approximately three times higher than in low water months, and sediment transported episodically [32,34,35]. Since the 1950s, sediment load in the Mississippi River has decreased to 145 MMT per year due to dams, bank revetments, meander cutoffs, other engineering structures, and improved soil conservation in the watershed [32]. The lower Mississippi River transports this water and sediment, and constructed a Holocene delta plain that forms ~30,000 km$^2$ of coastal Louisiana [33].

The Mississippi River Delta is a river-dominated delta building into the northern Gulf of Mexico receiving basin, which has relatively low wave and tidal energy [3,36]. Construction of the Mississippi River Deltaic Plain (Figure 1) occurred in a cyclic series of events, with major distributaries producing delta lobes within a delta complex on a ~1000–2000-year timeframe [33]. After depositing sediment and building a delta lobe, the river lost hydraulic efficiency due to a longer river course through the newly built delta, and switched to a more efficient route to the sea, starting the process of building another delta lobe [33,35].

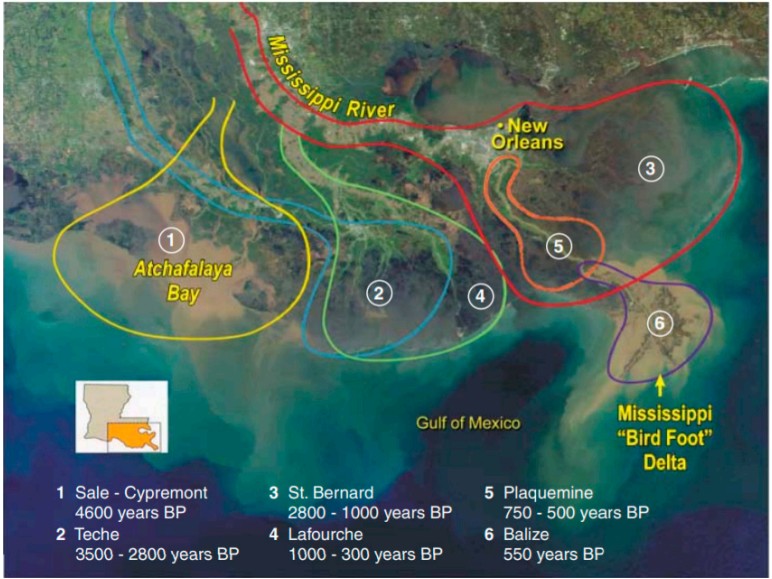

**Figure 1.** Mississippi River Delta Holocene history of delta growth (from [16], modified from [33]).

This cyclic delta building produced at least six major delta complexes [33,37] (Figure 1). The cyclical delta lobe building and degradational events cause coastal land building and retreat to occur at different places along the coast and times, although over the past ~7000 years a net building occurred [35]. The Chenier Plain in southwestern Louisiana (Figure 4) was also created by sediments from the Mississippi River transported westward [38]. The abandoned lobe enters a longer destructional phase where subsidence and marine processes dominate, leading to its deterioration. Wetlands on delta lobes can be sustained for a portion of the destructional phase through input of flood or storm derived sediments, and organic accretion [39–41]. The lobe also experiences marine reworking, with mouth-bar sands developing into barrier islands, which evolve into submarine shoals [42]. The combined processes of subsidence and marine reworking lead to the eventual formation of a coastal bay. The thick Holocene layer of the Lafourche delta, located in an entrenched alluvial valley, experiences high rates of subsidence, and, as predicted by Coleman et al. [35], the lobe appears to be undergoing a more rapid destructive phase than the St. Bernard complex [43]. The Atchafalaya River distributary currently captures ~30% of the Mississippi River flow via the Old River Control Structure, and is building the newest, Atchafalaya/Wax Lake delta lobe [44,45] (Figure 1 and Figure 4).

For over a century, coastal Louisiana has been the focus of intensive studies by geoscientists, biologists, engineers, and increasingly by social scientists and restoration practition-

ers due to its economic and ecological value, and its unique coastal communities [35,46,47]. These studies provide a framework for understanding the natural processes that the Mississippi River delta and Louisiana coastline will experience.

*2.2. Restoration in the Anthropocene: Situation and Need*

Historical episodic river flooding produced a deltaic landscape abundant in natural resources. However, the episodic floods were devastating to early European settlers, and river levees started to be built in the early 18th century to contain the river in its low flow channel during high flows, benefitting navigation, and reducing the risk of flooding of alluvial valley inhabitants, agriculture, and infrastructure. The principal management of the river for flood risk reduction and navigation has severed the river from its delta, and contributed to the severe land loss that threatens communities, economic resources, and vibrant ecosystems and the services they provide [16,46,48–50].

The Mississippi River levees cause river water levels in the channel to rise during flood periods (as compared to the natural floodplains), and after the Great Mississippi River Flood of 1927, the Flood Control Act of 1928 incorporated spillways and control structures into national flood control. Congressional appropriations in 1963 provided funding to build the Old River Control Structure floodgate system to regulate flow down the Atchafalaya River to 30% of the flow in the Mississippi River. This set of floodgates prevent further capturing of Mississippi River flow by the Atchafalaya River and limits building of the next Atchafalaya/Wax Lake delta lobe. Climate-related forcings have affected the opening of flood control structures such as increased operation of the Bonnet Carré Spillway (Figure 4); the spillway was opened an unprecedented two times in 2019, a historic high-water year fed by high rainfall throughout the Mississippi River Basin. Through project implementation, over 340 infrastructure projects such as barge gates, closure structures, crevasses, culverts, floodgates, locks, pumps, and weirs have been incorporated into the coastal Louisiana landscape, increasing the complexity of restoration planning.The region experienced an approximate 500-year flood in 2016 [51], and the Gulf of Mexico region experienced 8 major storms in 2020, including 3 hurricanes and 2 tropical storms that made landfall in Louisiana. These additional stressors provide logistical, financial, and planning constraints on communities and restoration planners who are also responsible for hurricane-related preparedness.

The Atchafalaya and Mississippi Rivers have a highly engineered watershed. Nutrient over-enrichment and hypoxia threaten resources and ecosystem services in the Gulf of Mexico [52], and are therefore among the most pressing environmental issues in the United States [53]. Oil and gas activities contribute to localized subsidence and wetland loss, and canals and navigation channels produce direct wetland loss, as well as indirect effects of saltwater intrusion and alteration of hydraulic flow and tidal patterns [54,55]. Invasive species, faulting, and storms can also contribute to land loss [56,57]. High rates of relative sea level rise increase the vulnerability of deltas, with rapidly-deposited sediments that compact and dewater leading to high rates of subsidence. Coastal Louisiana experiences some of the highest subsidence rates worldwide [58].

Overlain on the natural deltaic processes, anthropogenic infrastructure and processes now dominate the system in most places [59,60]. The ecosystem being restored in Louisiana is a "working landscape" where the natural and socio-economic systems are closely linked, making restoration management inherently complicated. Adaptations, and feedbacks of adaptation on the evolution of the delta, are hard to predict and project for planning purposes. Restoration of coastal habitats to replicate any historical coastal footprint is not feasible. Since 2007, CPRA has used over 165 million cubic yards of dredged sediment on restoration and protection projects, benefitted over 47,600 acres of land, constructed 60 miles of barrier islands and berms, and improved 340 miles of levees. As of April 2021, CPRA has 93 active projects that were identified through programmatic planning efforts, with 33 projects in construction, 51 in engineering and design, and 9 in the planning phase. Twenty-seven of these projects are hurricane risk reduction, 4 barrier island/headlands,

1 freshwater diversion, 3 hydrologic restoration, 40 marsh creation, 1 oyster barrier reef, 6 recreational use, 3 sediment diversion, 2 shoreline protection, and 6 other projects [61]. Total proposed project expenditures for state fiscal year (1 July through 30 June) 2022 are $887 million for 7 projects in planning, 37 in engineering and design, 66 in construction, and 161 in operation, maintenance and monitoring phases (Appendix A). Effective management of this complex deltaic coastal ecosystem and implementation of the ambitious restoration program requires reducing uncertainty in decision making and continual learning and improvement.

**3. Addressing Uncertainties in Programmatic Decision Making: The Louisiana Coastal Master Plan**

Hurricanes Katrina and Rita in 2005 converted coastal land to open water [15], caused the deaths of approximately 1200 people [62], and destroyed or majorly damaged more than 350,000 homes [63]. The devastation of Hurricanes Katrina and Rita triggered a recognition that coastal restoration and risk reduction, which at that time was implemented across multiple Louisiana state agencies, needed to be integrated and comprehensive. The federal government requested a central authority in Louisiana for all activities and funds, and development of a coordinated plan of action. The Louisiana State Legislature passed Act 8 in 2005, forming the Louisiana CPRA and mandating the agency develop and implement a comprehensive coastal risk reduction and restoration master plan, with regular updates every 5 years (now every 6 years) to account for the latest science and engineering, and a reevaluation of the goals and objectives. *Louisiana's Comprehensive Master Plan for a Sustainable Coast* (Coastal Master Plan) and its updates have become the State's principal programmatic blueprint for achieving a sustainable coastal zone.

The Coastal Master Plan builds on the long history of research to quantify land loss rates, patterns, and mechanisms [40,57,64–68], and previous planning efforts in Louisiana [22,69–71] (Appendix B). It is designed to provide the vision and direction needed to implement effective projects to meet the overarching goals to build land and reduce risk in coastal Louisiana. The Coastal Master Plan process informs management decisions on the programmatic scale, including:

- Which restoration and risk reduction projects to implement;
- How to most effectively use limited resources (e.g., funding, sediment);
- Grouping of projects and timing of implementation.

The objectives of the Coastal Master Plan are to improve flood risk reduction for communities and businesses, harness the natural processes that built Louisiana's coastal landscape, preserve coastal habitats and sustain the unique cultural heritage, and ensure the continuation of the commerce and industry of the working coast. The Coastal Master Plan is science based and developed using a participatory process, with input and engagement from diverse stakeholders in coastal Louisiana, as well as those with national interests.

*3.1. Science-Based Numerical Modeling Framework*

The scientific core of the iterative Coastal Master Plan development process is a linked suite of numerical models that project the response of the physical, biological, and human social coastal environment to proposed future restoration and risk reduction actions, including taking no action. Given the spatial scope of the Coastal Master Plan and the breadth of resources in Louisiana's coastal zone that it governs, that suite includes individual, but linked, numerical models that project:

- Hydrodynamics and water quality;
- Emergent wetland geomorphology and plant ecology;
- Barrier island geomorphology;
- Focal fish and wildlife species responses;
- Waves and storm surge;
- And human asset damage and risk assessment.

Building on previous restoration planning efforts in Louisiana, CPRA developed the 2007 Coastal Master Plan using a refined landscape change model originally developed for the Louisiana Coastal Area (LCA) Comprehensive Ecosystem Restoration Study (Appendix B). That refined model was used to predict the effects of the 2007 Coastal Master Plan project alternatives on the landscape [71].

A number of advancements were made to the predictive models for the 2012 Coastal Master Plan update, including incorporating feedbacks among various model components such as vegetation and hydrology [70]. Just a few months after the completion of the 2012 Coastal Master Plan, a 2017 Model Improvement Plan was developed that included recommendations based on external technical peer review of model components, input from the 2012 Coastal Master Plan modeling team, and input from two workshops with local, national, and international experts. The integration of the models used in the 2012 Coastal Master Plan was a large component of the improvements recommended in the 2017 Model Improvement Plan.

This resulted in an integrated framework named the Integrated Compartment Model (ICM) that includes natural processes that drive coastal land and ecosystem change, and considered anthropogenic infrastructure such as levees on the landscape. The ICM projects landscape and ecosystem changes based on relationships modeled for wetlands processes and vegetation, eco-hydrology, and barrier island morphology. A planning-level model, the ICM is computationally efficient for running 50-year coast-wide simulations to estimate candidate project performance under a range of future environmental conditions. Key ICM outputs include salinity and stage, land-water interface and elevation change, and changes in wetland vegetation.

New approaches to barrier island morphology and fish and shellfish community modeling were incorporated for the 2017 update. Improvements were also made to advance a more process-based approach to simulating physical and ecological dynamics, including process-based sediment distribution modeling. Model boundary condition datasets were updated and new stations added where available. An uncertainty analysis of the ICM investigated the uncertainty in key model outputs driven by uncertainties in model variables. Habitat suitability index (HSI) models underwent numerous improvements including development of new statistical relationships for key species, and the inclusion of new indices for blue crab and brown pelican. Recommendations for the various model improvements proposed or added for 2017 underwent an independent external review process and final reports were published by CPRA as technical appendices to the 2017 Coastal Master Plan [72].

The models used for risk assessment evaluate the effects of projects on storm surge and wave heights from tropical storms, and identify flood depths associated with different frequencies of inundation across the coast. The model geometry in selected areas was revised for the 2017 Coastal Master Plan update, and the model was validated with data from Hurricanes Gustav and Ike [73]. The Coastal Louisiana Risk Assessment (CLARA) model was modified to incorporate a larger floodplain, improvement of the inventory of coastal assets at risk, and updates to levee fragility scenarios.

Continuing the updating cycle, while the 2017 Coastal Master Plan is currently in place [22], the next step in the Coastal Master Plan process is currently underway with the development of the 2023 Coastal Master Plan. Recommendations from the Predictive Models Technical Advisory Committee, the Science and Engineering Board, and comments from external review of 2017 Coastal Master Plan components not addressed in the 2017 Coastal Master Plan are providing the basis for improvements to the 2023 Coastal Master Plan. Improvements are categorized as focused on predictive modeling by model type, the overall planning framework, and documentation/outreach/engagement. The list of improvements for the predictive modeling range from items as straightforward as code debugging to complex changes that would require restructuring of the modeling analyses from a deterministic to a probabilistic framework.

The predictive landscape, storm surge/waves, habitat suitability indices, and risk assessment models are being updated and refined, and newly available data are being used to update and refine the environmental and risk scenarios. New environmental data include longer datasets from the Coastwide Reference Monitoring System (CRMS)-*Wetlands* monitoring network, updated sea level rise projections, new subsidence data, and the National Structure Inventory asset inventory. There have been two solicitations for new project ideas from stakeholders; and outreach and engagement efforts are also proceeding. The 2017 Coastal Master Plan process showed that challenges facing coastal populations, businesses and ecosystems require a regional-scale response, and the 2023 Coastal Master Plan is integrating regional approaches to restoration and risk reduction.

### 3.2. Development of Future Environmental Scenarios Based on Updates to the State of the Science

A scenario approach is used for the Coastal Master Plan process to aid in decision making under uncertain future environmental conditions. Environmental drivers for the 2017 Coastal Master Plan scenario analysis were determined based on a review of relevant drivers used in the 2012 Coastal Master Plan analysis [74], a literature and data review to update plausible ranges of each driver, and an analysis of the impact on key model outcomes. To select the values for the drivers used in scenarios, the response of land area and landscape change projected by the ICM to environmental driver values was analyzed. Three environmental scenarios were selected (Figure 2), and outcomes for project performance over time were projected across this range of possible future conditions.

| SCENARIO | SEA LEVEL RISE | SUBSIDENCE | STORM FREQUENCY | AVG. STORM INTENSITY | PRECIPITATION | EVAPO-TRANSPIRATION |
|---|---|---|---|---|---|---|
| LOW | 0.43 m | 0 to 19 mm/yr; varies spatially | -28% change of current frequency | +10.0% of current central pressure deficit | >HISTORICAL | <HISTORICAL |
| MEDIUM | 0.63 m | 0 to 19 mm/yr; varies spatially | -14% change of current frequency | +12.5% of current central pressure deficit | >HISTORICAL | HISTORICAL |
| HIGH | 0.83 m | 0 to 25 mm/yr; varies spatially | 0% change of current frequency | +15.0% of current central pressure deficit | HISTORICAL | HISTORICAL |

**Figure 2.** Environmental scenarios used in the 2017 Coastal Master Plan.

The definition of the component environmental scenarios provides the basis for the initial modeling framework runs, which assume no further restoration or risk reduction actions are taken in the future. A no further action approach can be considered as a management option, and then serves as the point of comparison for model runs with additional projects implemented.

### 3.3. Candidate Projects Identified through a Participatory Approach

New candidate projects were considered in the 2017 Coastal Master Plan. A participatory approach to the identification of new projects was undertaken with the New Project Development Program (NPDP), and new project ideas solicited by CPRA during two solicitation periods. Restoration project types include hydrologic restoration, shoreline protection, bank stabilization, oyster barrier reef, ridge restoration, marsh creation, and barrier island restoration. Risk reduction project types include structural (e.g., earthen levees, T-walls, and floodgates) and nonstructural risk reduction (e.g., non-residential floodproofing, residential elevation, and residential voluntary acquisition). Restoration and

protection projects were evaluated for their effectiveness in building and maintaining land and reducing risk, and suites of projects (alternatives) were evaluated both for ecosystem benefits and reduction of levels of risk.

A planning tool that uses elements of multi-criterion decision analysis and robust decision making was used in the 2017 Coastal Master Plan to facilitate science-based planning level decision making. The predictive models were used to evaluate the coastal restoration and risk reduction candidate coastal projects individually, and as groups of projects, or alternatives. The planning tool incorporated results from the 2017 Coastal Master Plan predictive models as inputs, as well as including planning constraints such as sediment availability and funding (Appendix A), and stakeholder preferences. Results of the predictive models and the planning constraints were used by the planning tool to develop several project alternatives based on optimization to maximize the goals of land building and risk reduction under environmental scenarios. Stakeholder input was incorporated by evaluating the sensitivity of project selection to metrics such as brown and white shrimp habitats.

### 3.4. Coastal Master Plan Updates Informed by Scientific Approaches

While the Coastal Master Plan effort is thus informed by the objective numerical analyses informed by the state of the science, it is also responsive to the needs of communities. Decision making for the 2017 Coastal Master Plan was informed by the public, and CPRA engaged with an expanded group of stakeholders to better understand challenges citizens and communities face from coastal land loss, provide information to stakeholders, and improve the integration of local and state activities. CPRA hosted meetings across the coast to engage communities, from small groups to larger open house meetings that facilitated interactions and opportunities to receive input. Technical briefings were also hosted to provide information about the technical approach, and to receive feedback. Four public hearings were held in an open house format that included informative exhibit booths, a presentation, public comments, and informal conversations with CPRA staff.

Advisory groups provided important recommendations and guidance to inform the 2017 Coastal Master Plan development process. Nationally- and internationally-known scientists, engineers, and planners provided working-level guidance via the Predictive Models and Resiliency Technical Advisory Committees, and technical review and recommendations via the Science and Engineering Board. Members of the Framework Development Team provided insight and counsel, and included representatives from federal, state, and local governments, NGOs, business and industry, academia, and the coastal community. Five focus groups integrated community, fisheries, landowner, energy, industry, and navigation perspectives about restoration and risk reduction projects.

The Coastal Master Plan models and decision support tools were used to project the performance of projects or project alternatives, and compare projections to Future Without Action results. The 2017 Coastal Master Plan recommends 124 coastal risk reduction and restoration projects that are projected to reduce expected flood damages by $150 billion and build or maintain 2077 km$^2$ of land over 50 years (Figure 3). These results are obtained from a comparison to Future Without Action under a medium environmental scenario (Figure 2). Funding of $25 billion is allocated for nonstructural and structural risk reduction projects, and $25 billion for restoration projects. Underscoring the need for using natural processes to build and sustain coastal land, the 2017 Coastal Master Plan, like the 2012 Coastal Master Plan, identified eight proposed sediment diversions that provide long-term benefits.

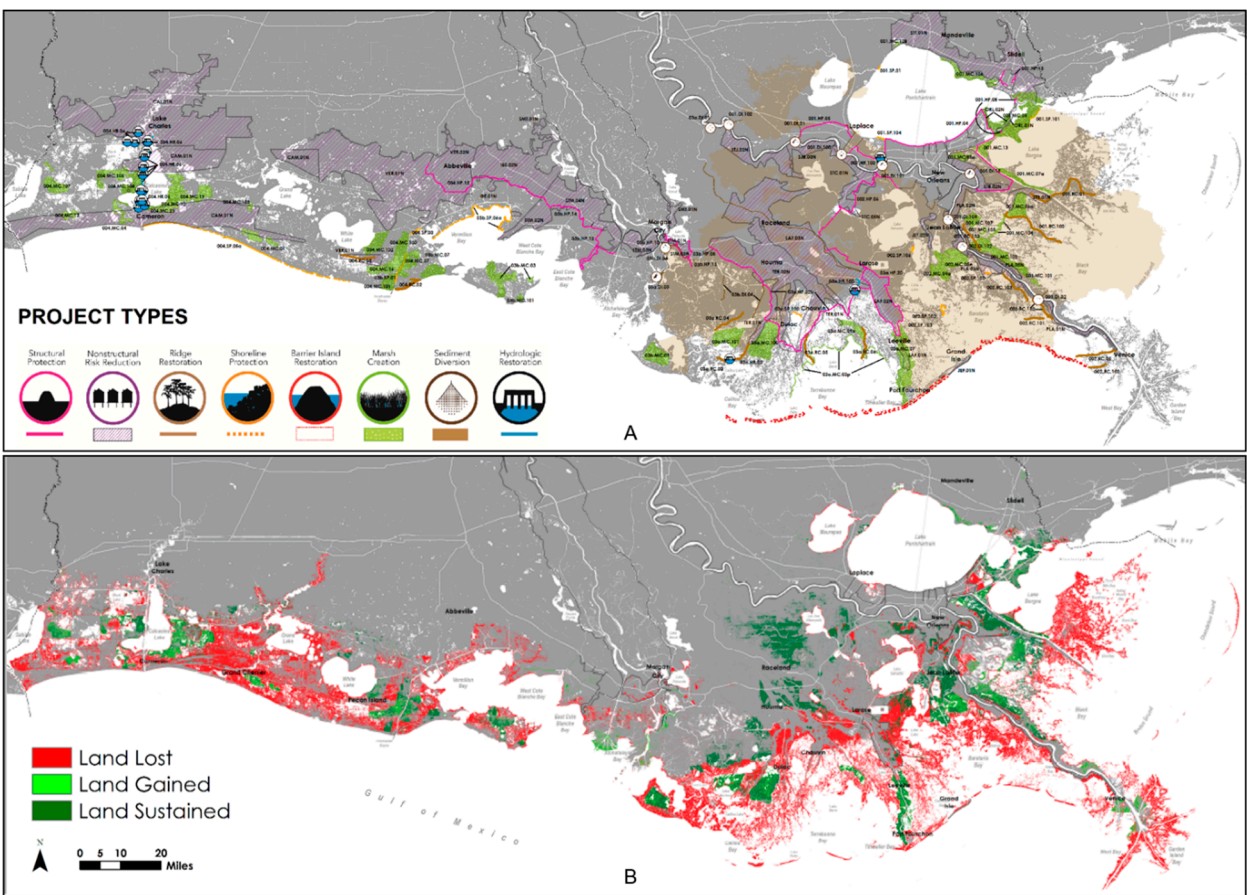

**Figure 3.** (**A**) The 2017 Coastal Master Plan recommends 124 projects, with a projected $150 billion in flood damage reductions for the medium environmental scenario at year 50 [22]. (**B**) Land lost, gained, and sustained with the 2017 Coastal Master Plan for the medium environmental scenario, with a projected 2077 square kilometers of land gained or sustained at year 50.

### 3.5. Lessons Learned in Addressing Uncertainties in Programmatic Decisions

*External and internal input refines model improvement:* the Model Implementation Plan is critical for providing guidance to the Coastal Master Plan updating process. Input from both external and internal experts improves robustness and incorporates successful techniques from other restoration programs into strategic planning and implementation.

*Scenario approach provides insight in uncertain environmental conditions:* substantial uncertainties remain in future environmental conditions, such as related to climate change. The scenario approach enables these uncertainties to be addressed by exploring the effects of a range of possible future conditions, and how these environmental conditions may affect project performance over 50 years. The Future With Action prediction clarified that even with complete plan implementation, there was still a net loss of 3761 km$^2$ projected for the Medium Environmental Scenario over 50 years. The Future Without Action projections provide a baseline to compare implementation of programs and projects to build land and reduce risk.

*Usefulness of landscape-scale analysis for programmatic decisions:* the Coastal Master Plan landscape-scale analysis process allows projection of future locations of land, water, vegetation, and productive fish and wildlife habitat, as well as flooding from storms across the entire Louisiana coast. Utilizing a reduced complexity model such as the ICM to represent coastal hydrology allows for reduced simulation times compared to other numerical models. For instance, a 50-year simulation in the ICM takes approximately 10 days, whereas a similar duration simulation of one-third of the coastal zone domain would take approximately four times that amount of time with a model that resolves coastal

hydrology with a higher fidelity. This provided insights for implementation, including that projects often provide minimal benefits beyond their immediate project footprint. However, synergistic interactions among projects of different types affecting the same region can produce greater and longer-term benefits.

*Importance of regular updates:* advancements in the algorithms, grid, and integration of the 2017 Coastal Master Plan Integrated Compartment Model, and the incorporation of new data refined projections of project performance under uncertain future conditions.

*Effectiveness of harnessing natural system processes:* the Future Without Action projected that 8838 km$^2$ would be lost in 50 years under the Medium Environmental Scenario, and 10,679 km$^2$ under the High Environmental Scenario [22]. Most of coastal Louisiana is expected to experience land loss without additional projects, except in areas where the natural deltaic land processes are still existent such as in the Atchafalaya and Wax Lake Deltas.

*Decision support tool facilitated science-based decision making*: the decision support tool enabled identification of a final plan based on science that most effectively reached the goals of building land and reducing risk.

*Uncertainties in boundary conditions are likely greater than model uncertainties*: using model performance statistics and a suite of environmental boundary conditions representing possible future climate change and relative sea level rise scenarios, uncertainty in future landscape configurations was analyzed with the ICM [75]. This analysis focused on spatiotemporal patterns of land change and found that while magnitude of uncertainties varies over time and space, the uncertainty attributable to model performance errors is most impactful on overall uncertainty under the least severe relative sea level rise assumptions. As higher rates of relative sea level rise are assumed into the future, the impact of model errors is overwhelmed by the impact of relative sea level rise.

*Frequent stakeholder outreach critical:* the robust stakeholder outreach for the Coastal Master Plan informs both stakeholders and CPRA, and is an important component of effective management. The plan is supported both inside and outside Louisiana; the Louisiana State Legislature unanimously approved the 2007 Coastal Master Plan, and the 2012 and 2017 Coastal Master Plan updates.

## 4. Monitoring and Adaptive Management to Assess Project and Program Performance

*4.1. System Wide Assessment and Monitoring Program*

Data collection and information management are key components of CPRA's technical processes, providing necessary data and information on program and project performance to decision-makers on a timely basis. Monitoring and Adaptive Management inform management decisions, including:

- Whether program or project implementation should be altered based on performance;
- Whether project operations should be changed due to environmental conditions.

The System-Wide Assessment and Monitoring Program (SWAMP) has been developed to monitor and assess both natural and human systems in coastal Louisiana to ensure a comprehensive network of coastal data collection activities is in place to support the development, implementation, and adaptive management of the coastal risk reduction and restoration program within coastal Louisiana [76]. The focus of SWAMP is to obtain repeated long-term (e.g., years to decades) measurements that can be analyzed to detect changes that may result from both small- and large-scale restoration and risk reduction projects, environmental disturbances, and other major drivers that impact the system.

SWAMP is being designed in a nested fashion to facilitate the integration of project-specific data needs into a larger, system-level design framework. It encompasses existing monitoring programs including the Coastwide Reference Monitoring System (CRMS)-*Wetlands*, a network of ~390 sites in coastal Louisiana where a wide variety of empirical wetlands data are collected, and the Barrier Island Comprehensive Monitoring (BICM) Program, first initiated in 2007 by the State to monitor all of the barrier islands and barrier shorelines in Louisiana. SWAMP water quality monitoring, geophysical surveys,

subsidence studies, sediment budgets, hydrology, above and belowground vegetation biomass, and fisheries monitoring have been implemented across much of the State's coastal area (Figure 4).

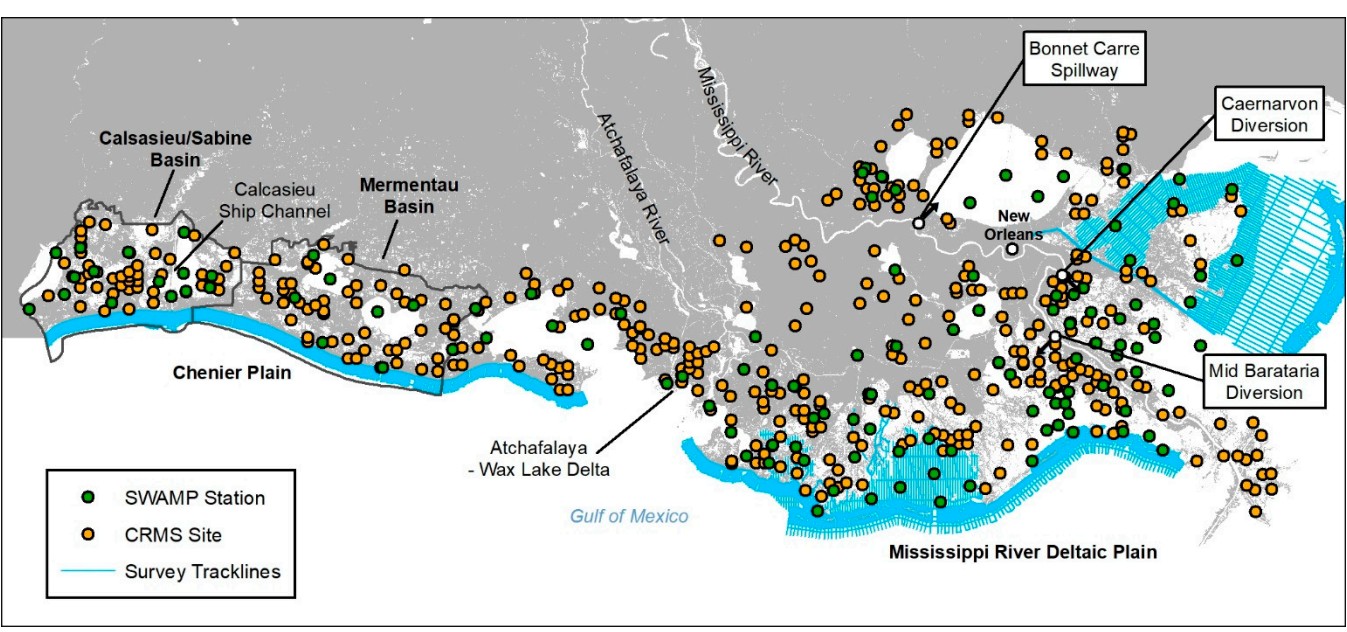

**Figure 4.** System-Wide Assessment and Monitoring Program above and belowground vegetation biomass, water quality, hydrology, and geophysical survey monitoring implementation.

### 4.2. Adaptive Management Framework

Adaptive management has been used in natural resource management to reduce uncertainty for more than four decades [77,78]. Adaptive management facilitates resilient policies for managing ecosystems with the basis of using the best available knowledge to design and implement management plans, while establishing an institutional structure that enables learning from outcomes to adjust and improve decision making. Adaptive management procedures have long been incorporated into coastal Louisiana restoration processes [71,79,80]. Louisiana's complex and dynamic coastal area, uncertainties in future environmental conditions, and difficulties in predicting outcomes of restoration and risk reduction projects all present challenges that benefit from an adaptive management approach [81].

The Coastal Master Plan is developed in an adaptive management framework, with the recurring update cycle providing the basis for incorporating new scientific and system dynamics, and project performance information into decision making [82]. By allowing flexibility in implementation as conditions change, CPRA's Adaptive Management program is essential to the long-term performance of projects and the achievement of the greatest amount of positive ecosystem improvement. Projects selected for implementation by the Coastal Master Plan process that still have substantial uncertainties undergo feasibility investigation to analyze alternative project siting and operational details, and to begin developing Engineering and Design details beyond those established during the Master Plan evaluations. Pending an outcome of project feasibility, and once 30% design of the project is reached, the project moves to a more formal Engineering and Design phase, where 100% design is attained. Once constructed, a project is monitored and, if appropriate, adaptively managed, including its operations and maintenance. New information and lessons learned from project monitoring are then incorporated into the next iteration of the Coastal Master Plan (Figure 5).

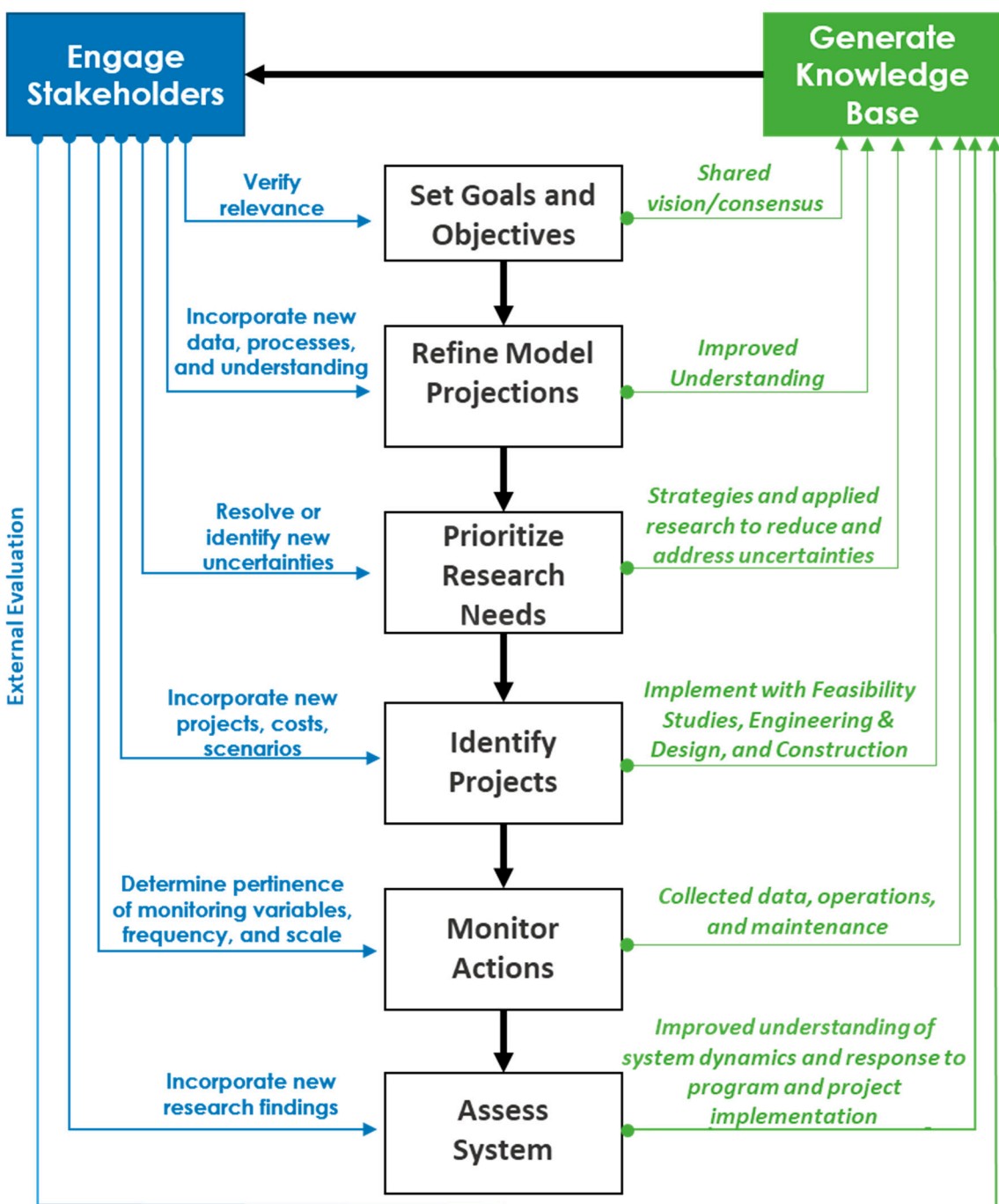

**Figure 5.** Program and project implementation process used for Louisiana coastal restoration demonstrating how science is incorporated to reduce and address uncertainties. Modified from [82].

*4.3. Lessons Learned in Addressing Uncertainties in Monitoring and Adaptive Management Decisions*

*Adaptive Management framework supports decision making*: CPRA believes that not taking action due to uncertainty is not an option, since the Future Without Action is the loss of large portions of the deltaic plain due to sea level rise, subsidence, and other factors. An Adaptive Management framework supports action urgently needed to reduce land loss and risk by reducing the uncertainty of implementation action compared to the Future Without Action.

*Monitoring must align with program and project goals and objectives*: The development and implementation of SWAMP provides much of the data that will be used to evaluate

and manage the overall risk reduction and restoration program and also serves as the backbone of project-related monitoring needs. Monitoring and operation of restoration and risk reduction projects are nested within a larger basin-wide and coast-wide SWAMP framework and will allow informed decisions to be made with an understanding of system conditions and dynamics at multiple scales. It is important that program and project goals and objects are complementary to and align with the long-term monitoring program, and are tracked at many scales.

## 5. Improving Knowledge of System Dynamics and Response to Program and Project Implementation

An important part of the CPRA's Adaptive Management framework is the identification and resolution of uncertainties to inform future decision making. Louisiana's dynamic coastal environment and shifting baselines associated with ongoing landscape change results in the need to both identify and prioritize research needs and synthesis efforts that fill information gaps and assess project and program progress. Applied research and science synthesis is a knowledge base component of CPRA's Adaptive Management process, and helps inform management needs at the project, regional, or coast-wide scales including:

- Improved understanding of system dynamics;
- Improved predictive capabilities of system response to management actions;
- Assessment of system's response to management action;
- Assessment of the performance of project types.

Information from project-specific and other research and synthesis activities are shared both internally and with external audiences in reports, and along with data, models, maps, and other deliverables are all publicly available in CPRA's Coastal Information Management System [83].

### 5.1. Identifying Science Needs

Components of CPRA's research and synthesis efforts include a process for identifying, compiling, prioritizing, and addressing information through targeted studies (research) or through data and information aggregation (synthesis). Research needs are identified by personnel from all divisions of CPRA, literature compilation and review, input from external entities (e.g., science advisory panels, technical workgroups, and researchers), the identification of information gaps during project or program development, CPRA project monitoring reports, and responses to specific events.

Priority research areas for CPRA include:

- Understanding uncertainties in project engineering and design, implementation, and sustainability;
- Reducing uncertainties in future environmental conditions;
- Improving data and assumptions used in monitoring and assessment, predictive models, and decision-support tools;
- Understanding the dynamics of the social, environmental, and economic coastal system, and the effects of land loss and implementation of the Master Plan in these systems; and understanding social, cultural, and economic resilience, and the adaptability of coastal communities to natural disasters and long-term land loss.

### 5.2. Collaboration with Gulf-Wide and Regional Efforts

The State of Louisiana and its federal and Gulf of Mexico state partners have allocated considerable resources and have made long-term commitments to the restoration and management of wetland and aquatic resources in the Gulf of Mexico coastal zone. CPRA is sharing institutional knowledge and capacity, and learning from other Gulf states and entities in a number of regional restoration efforts (Appendix C). Working groups and entities established after the 2010 Deepwater Horizon Oil Spill that include representatives from all five Gulf of Mexico states have fostered unprecedented coordination and collabo-

rative working relationships. CPRA also benefits from, and is engaged in, project advisory committees for multiple external funding programs to increase the relevancy of research projects to Louisiana's coastal management decisions.

*5.3. Using Science to Inform Feasibility Decisions and Refine Project Implementation of River Diversions*

For certain project types, development funds are dedicated for feasibility studies and design phases, wherein the project undergoes additional development that includes exhaustive analyses and multi-agency collaboration (Figure 5). Complex restoration or risk reduction projects types (e.g., diversions) have funds available to support research that addresses important project-specific information gaps or uncertainties. A key component of the project implementation process is to more fully investigate technical uncertainties to maximize the benefits of these projects while minimizing unintended consequences.

Nowhere within CPRA's project portfolio is the need for multiple avenues of applied research as necessary as with the sediment diversion projects outlined in the Coastal Master Plan. Multiple scientific studies have attributed at least part of the State's coastal wetland loss since the 1930s to the isolation of the coastal wetlands from sustaining freshwater, nutrients and sediments due to the construction of the Mississippi River and Tributaries flood protection levees following the 1927 Great Mississippi River Flood. A cornerstone method to restore and protect the Mississippi River Delta using natural processes is to reconnect the river to its delta by diverting Mississippi and Atchafalaya River flows into the State's coastal wetlands and open water bodies [22,84–86]. The proposed sediment diversions are intended to divert freshwater and sediment from the Mississippi or Atchafalaya rivers into adjacent coastal wetlands in an effort to restore land-building processes that were interrupted by the construction of levees on the river and to lessen the trend of land loss that has plagued coastal Louisiana [22]. Lessons learned from implementation, operation, and assessment of existing freshwater diversions in the State inform additional sediment diversion planning and design.

The first of these projects that the State is developing is the Mid-Barataria Sediment Diversion, with an estimated cost of $1.4 B (Figure 4). This sediment diversion project was included on the Federal Permitting Dashboard in 2017, a government-wide effort to streamline federal permitting and review, and increase transparency [22]. CPRA has conducted feasibility and engineering and design analyses that projected the Mid-Barataria Sediment Diversion would create and sustain 28 square miles of land, and the project's draft Environmental Impact Statement (EIS) was recently released for public comment by the U.S. Army Corps of Engineers in March 2021. The analyses supporting the DEIS occurred over a period of several years, owing to the scale of the proposed project (a maximum proposed discharge of 2124 cubic meter per second (cms) flow from the Mississippi River when river discharge is at 28,317 cms and the concomitant scale of the inferred physical, ecological and social effects of those flows on the Barataria Basin. Not only have substantial resources been devoted to the development of a suite of high-resolution numerical models needed to project those receiving basin responses, State and federal partners have relied on a long list of scientific investigations on estuarine resources under historical and current conditions, and the potential effects of freshwater and nutrients on those resources under a potential Future With Action.

Recently, in recognition of the uncertainties surrounding diversion projects, CPRA identified the need to synthesize and document the effects of diversion-borne freshwater, nutrients and sediments on receiving basin wetlands and estuarine water bodies, since scientific differences in opinion surrounded the potential for beneficial and detrimental outcomes from implementing river diversions. CPRA scientists partnered with leading experts in the field of estuarine and coastal wetland ecology to synthesize the effects of diversions-borne freshwater, nutrients, and sediments on receiving basin wetlands and estuarine water bodies to account for recent literature and relevant SWAMP data. This working partnership allowed incorporation of wider community expertise to inform strategic decision making. Results were published in a virtual Special Issue in *Estuarine, Coastal*

*and Shelf Science* that synthesized aspects of changes in deltaic wetlands and open water bodes that would occur in the absence of restoration action as well as in response to river diversion included in the 2017 Coastal Master Plan [87]. CPRA has incorporated information from the synthesis into nutrient-effect model development for diversion projects such as the Mid-Barataria Diversion Project and will integrate that same information into subsequent Coastal Master Plan and project-specific planning models to refine modeling of ecosystem response to riverine nutrients.

CPRA and its federal partners have since drafted an extensive Project Monitoring and Adaptive Management Plan for the Mid-Barataria Sediment Diversion, which outlines a robust data collection and analysis effort intended to occur well into the future. That effort will provide a large body of empirical data to validate the numerical models used for project planning, and support efforts to confirm, refute, or otherwise amend many of the scientific theories which currently underpin the agency's diversion program.

*5.4. Research to Resolve Uncertainties*

5.4.1. Coastal Science Assistantship Program

The Coastal Science Assistantship Program (CSAP) directs scientific research to answer questions about planning, designing, constructing and evaluating coastal risk reduction and restoration projects. CPRA awards, which fund a three-year period of study for Master of Science students enrolled full time at Louisiana colleges/universities, are intended to support academic research that will improve the planning, design and construction of coastal restoration projects, thus contributing to CPRA's overall success. In addition, these assistantships help CPRA foster closer ties with the academic community and promote a platform for collaboration by developing relationships with students and professors. These improved relationships allow for greater communication and participation in the State's coastal risk reduction and restoration program. In addition to monetary support for up to three years, participation in the CSAP provides students invaluable professional working experience beyond that gained in traditional academia. The required internship with CPRA staff offers on-the-job training that promotes understanding of CPRA's daily activities and of broader issues relevant to coastal risk reduction and restoration. The Louisiana Sea Grant College Program manages the contracting, with funding and internship supervision provided by CPRA. Research projects are selected by a team of CPRA personnel who evaluate each application for technical merit, originality, credibility, and relevance to CPRA activities. Since CSAP's inception in 2008, the program has supported 66 Master of Science students at Louisiana institutions. Since CSAP's inception in 2008, the program has supported 66 Master of Science students at Louisiana institutions, 49 theses and more than 9500 internship hours with CPRA. Students supported by this program have gone on to produce more than 90 journal publications.

5.4.2. RESTORE Act Centers of Excellence Research Grants Program

Louisiana will receive approximately $26 million over 15 years from the RESTORE Act Center of Excellence Research Grants Program to support research relevant to implementing the Coastal Master Plan through grant allocation. The Louisiana Center of Excellence has three advisory entities. The Executive Committee is comprised of senior research officials from Louisiana's universities and research organizations and is weighted towards those with a strong historic focus on coastal issues. The External Review Board is a group of independent scientists and engineers convened to provide technical feedback on an original Research Strategy and to serve as a panel for research proposal review. The Technical Working Group is composed of subject matter experts and worked with Center of Excellence staff to develop the original Research Strategy and a subsequent Research Needs document. As of the completion of final project reports by 2020, 40 peer reviewed publications were published or in press, over 60 conference presentation were completed or planned, and 46 undergraduate and graduate students participated in the research projects.

The value of the Center's relevant research is demonstrated by one of the LA-COE graduate studentship research projects selected for funding during Phase I of the RESTORE Center of Excellence Research Grants Program. The project evaluated radar-based precipitation datasets and the evolution of radar-rainfall performance [88]. This research was relevant to CPRA's knowledge base to better understand short-term precipitation patterns, and improve meteorological forcings on coastal system models. This project was also directly relevant to integrating radar-based precipitation products into the 2017 Coastal Master Plan ICM hydrology subroutine. The improved models for the 2023 Coastal Master Plan now use bias-corrected radar-rainfall timeseries, a distinct improvement over the 2017 modeling effort given the paucity of rainfall gages within the coastal zone of Louisiana [88].

*5.5. Documenting Progress and Resolving Uncertainties through Reports and Synthesis Efforts*

5.5.1. Coastal Wetlands Planning, Protection and Restoration Act Reports to Congress

Evaluation of the effectiveness of projects implemented is an important component of successful management. Congress established the Coastal Wetlands Planning, Protection and Restoration Act (CWPPRA) in 1990 (Public Law 101–646, Title III) in response to recognition of the ongoing severe coastal wetland losses in Louisiana and the increasing impacts on locally, regionally, and nationally important resources. Congress established and directed the Louisiana Coastal Wetlands Conservation and Restoration Task Force (Task Force) to prepare, annually update, and implement a list of coastal wetland restoration projects in Louisiana to provide for the long-term conservation of wetlands and dependent fish and wildlife populations. In addition, Congress directed the Task Force to provide scientific evaluation every three years on the effectiveness of the projects as required by Section 303 (b) (7) of CWPPRA and their benefit to fish and wildlife. CWPPRA collects information both at the project level and coastwide through CRMS to assess cumulative benefits of restoration.

The report typically provides an overview of the CWPPRA program and process, CWPPRA's benefits to fish and wildlife, information on the CWPPRA project selection process and planning and implementation, evaluation of the CWPPRA program with CRMS monitoring data, and evaluation summaries of selected CWPPRA projects. CWPPRA has served as the proving ground for many restoration techniques and a model for future projects yet to be designed. Many lessons learned through CWPPRA have been incorporated in current restoration projects (e.g., CWPPRA demonstration projects). Since its inception, CWPPRA has protected and restored almost 36,500 hectares (358 square kilometers) of Louisiana's coastal wetlands in its 1990–2015 projects.

5.5.2. Hydrologic Basin-Level Assessments and Coast-Wide Science Synthesis

Critical to successful management of the coastal Louisiana ecosystem are quantifying the baseline conditions and evaluating change in the natural and human systems to support the strategic implementation and assessment of restoration projects. Louisiana's coastal zone has historically been sub-divided into nine separate, functionally-distinct (for the most part) hydrologic basins. CPRA has initiated hydrologic basin-level natural systems assessments to synthesize information and data at the regional, or hydrologic basin level. These reports: (1) synthesize historical and current conditions; (2) summarize the constructed restoration and risk reduction projects within the basin; (3) assess the individual and cumulative effects of a project or suites of projects on the condition (e.g., localized land loss, hydrologic functioning), and restoration goals within the basin; and (4) improve the understanding of the collective effectiveness of restoration projects. Within-basin sources of variation are assessed, with a current primary focus on recent CRMS and SWAMP data within the basin. Recommendations are included to improve the outcomes of restoration and risk reduction implementation.

The value of the Basin-Level Reports was recently illustrated in altered CPRA decision making regarding the Calcasieu Ship Channel Salinity Control Measures Project (Figure 4), which is a hydrologic restoration project in southwestern Louisiana, selected

for the 2017 Coastal Master Plan's first implementation period, with an estimated project cost of ~$260 M. Hydrology of this Chenier Plain area has been substantially altered by navigation channels such as the Calcasieu Ship Channel. The 2017 Coastal Master Plan projected approximately 470 km$^2$ of land loss in the Calcasieu/Sabine Basin over the next 50 years under the Future Without Action. The CPRA basin report of the region found that present environmental, structural, and operational conditions have altered processes in the Basin such that persistent inundation and resulting flood stress is currently the main contributor to marsh vulnerability in this system, while salinity is currently better controlled than it was in previous decades [89]. CRMS monitoring data were instrumental in finding that both flood stress and high salinity pose critical threats to marsh health if not controlled. The original project was designed to reduce the rate of land loss by maintaining brackish and, farther from Calcasieu Lake, fresh salinity regimes longer to maximize organic marsh accretion, thus maintaining sufficient elevation to mediate sea level rise-driven collapse for as long as possible. Based on the findings in the basin report, the project underwent a reanalysis of projected project impacts, and CPRA concluded that the original project should be adjusted, with a path forward that includes a combination of large-scale marsh creation/nourishment and substantial marsh drainage improvements. This change of scope of the project from reducing the rate of land loss through salinity control to other measures that reduce flood stress such as marsh drainage improvements and marsh creation and nourishment, was a direct result of increased understanding at the basin scale that was incorporated to adjust a project-level management decision, and associated engineering and design activity and funding.

### 5.5.3. Project Monitoring Synthesis

The shift to basin-level data summaries does not mean that project-specific monitoring is not continuing Developing scientific synthesis to resolve knowledge and information gaps is important to the success of the risk reduction and restoration program. Monitoring programs for CPRA's restoration and risk reduction projects are currently being managed and/or conducted by CPRA and the United States Geological Survey. Project-specific monitoring often begins prior to construction in order to establish baseline data, and then continues once construction of projects is complete. Monitoring activities are conducted to determine how well a project is progressing towards meeting its goals and objectives. Monitoring reports are prepared and are available for review by the public on CPRA's website. Each report contains information gathered and analyzed by scientists and engineers describing operation and maintenance of the project and whether or not it is functioning as expected. Project-specific reports are written for all stages of project development, from feasibility studies, engineering and design, to monitoring and assessment. CPRA's monitoring programs provide CPRA the opportunity to evaluate the effectiveness of individual projects at multiple scales as well as the combined effects of multiple projects across ecological basins. Synthesis projects on specific topics that have scientific uncertainties are also undertaken.

### 5.6. Targeted Research Drives the Expansion of the Louisiana Coastal Knowledge Base

Over 150 research projects were funded by CPRA from 2010 through March 2021 (Figure 6), which do not include project-specific modeling studies or feasibility studies. CPRA-funded research projects have produced high-quality research results that are applicable to the overall (multi-agency) coastal program's knowledge base, and often directly applicable to project and program implementation. CPRA-funded research projects increased knowledge about processes and techniques relevant to planning efforts for numerous topical areas (Figure 6). Understanding geological mechanisms in the complex Mississippi River Deltaic Plan is important for predicting interactions of geological effects on projects, and is one of the largest uncertainties in modeling projections. CPRA funded work that increased understanding of subsidence rates in the coastal basins, faulting, consolidation and overwash processes, and coastal landform evolution, which reduced uncertainties in

geological components of program and project implementation. Geotechnical considerations are also important in project planning, and applied research projects characterized dredged sediment, settlement processes, and determined that sampling/specimen size does not have a significant effect on undrained shear strength in soft soils. Results are applicable to project planning that accounts for differences in soil, slurry, and marsh fill materials.

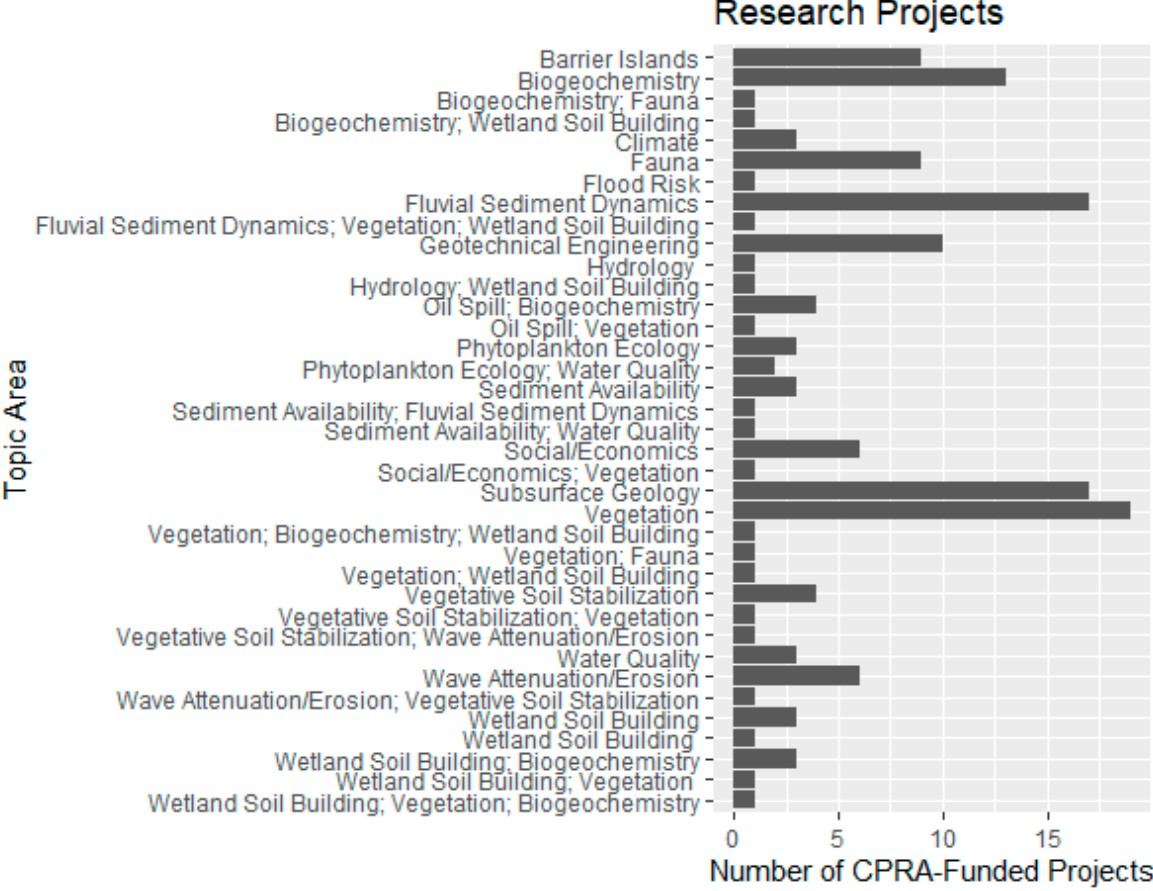

**Figure 6.** CPRA-funded research project topics from 2010 to March 2021. The projects do not include project-specific modeling studies or feasibility studies.

The effects of coastal restoration and risk reduction projects on coastal habitats and wildlife are also important considerations for project and program implementation. Funded research projects increased understanding of river diversions and nutrient influences on wetlands and soils and phytoplankton dynamics. Effects of diversions on salinities and fauna were clarified. Research also enhanced flood hazard assessment capabilities through development of coupled hydrologic and surge processes modeling. Increased understanding of the dynamics of the Mississippi and Atchafalaya Rivers, and estuarine gradients and mixing are key components of improved modeling and projection. Hydrograph-based sediment availability and investigation of river sediment dynamics in response to flow regulation and river engineering found impacts on sedimentation patterns and increased avulsion risk. Interactions among hydrology, sediment, and vegetation were characterized in the Wax Lake Delta, and for specific wetland species.

Social/economics research and the growing recognition of its importance to decisions on risk reduction and other projects, and in planning for population transitions, led CPRA to initiate funding in this topic area in 2013. Research increased understanding of how to integrate traditional knowledge from local stakeholders into decision making, and the economic implications of continued land loss. Research on designing and facili-

tating an equitable relocation strategy provided CPRA a conceptual model for relocation decisions [90].

Over 2800 technical reports developed since 1980 have documented institutional knowledge such as lessons learned, and synthesized the vast amounts of information and data collected (Table 1). Reports have also focused on specific management-related questions (e.g., evaluation of certain restoration techniques, model performance, and geotechnical settlement) [83]. Basin-level assessments summarized projects and cumulative effects of projects on restoration goals within the Mermentau [91] and Calcasieu-Sabine Basins [89] (Figure 4). Preliminary assessment of the Terrebonne Basin found that salinity and inundation drive land change (gain or loss) and vegetation richness patterns in the basin. Lessons learned have been documented and applied including improved design of barrier island projects based on performance [92], and refined project types authorized based on performance and improved understanding of the problems [93].

**Table 1.** Project, programmatic, and synthesis reports developed since 1980 in support of coastal Louisiana risk reduction and restoration efforts.

| Report Type | Number of Reports | Scope |
| --- | --- | --- |
| Data collection | 321 | Data collection methods, QAQC, summary |
| Survey | 465 | Baseline and project surveys and methodologies |
| Geotechnical | 111 | Geotechnical properties and engineering analyses |
| Feasibility | 162 | Design guidelines to assess project feasibility |
| Environmental assessment | 55 | Evaluation of environmental impacts |
| Project completion | 148 | Project features, benefits, personnel, construction costs |
| Ecological review | 48 | Review of project design as of 95% design review |
| Operations, maintenance, and monitoring | 459 | Project-specific operations, maintenance, and monitoring |
| Inspection | 424 | Project engineering and structural integrity |
| General programmatic | 636 | Annual plans, adaptive management bulletins, QAPPs, communication documents, other |

*5.7. Lessons Learned in Addressing Uncertainties in System Dynamics and Response to Program and Project Implementation*

*Value of a robust monitoring program:* data collected by CPRA's monitoring programs were important sources for updating coastal datasets used in modeling and assessment efforts.

*Formalize, expand, and support process to identify and prioritize research and synthesis needs:* identifying research priorities helps focus research funding on critical uncertainties for decision making. Coordination and collaboration in the identification and prioritization of research and synthesis needs across programs in Louisiana is needed to avoid duplication of efforts and to discuss opportunities of leveraging resources and maximizing efficiencies in addressing research and synthesis needs.

*Increase effective communication of research needs:* it can be very difficult to obtain research that addresses a critical management need. Researchers can be narrowly focused on their area of expertise, and determining research priorities to address management needs that are both specific to a policy or practice, and that contribute to general knowledge and understanding takes substantial effort. CPRA has shifted to targeted requests for proposals for research programs that identify priority research needs that are a relevant fit for the program. Through various review processes, CPRA has found that determining if a research proposal is relevant to program implementation is difficult if a reviewer is external to the agency. Additional strategic actions should be undertaken to ensure new, external, and relevant ideas are incorporated and communicated.

*Foster opportunities to synthesize project-specific reports:* basin-scale synthesis reports help assess effects of multiple projects, and inform management of projects to maximize synergistic effects. A centralized effort should be developed for synthesizing project-specific reports to address broader information and research needs, and provide consistency for aggregation and evaluation of project performance. Expanded efforts to synthesize

aspects of changes in deltaic wetlands and estuarine landscapes that could occur in the absence of restoration action as well as in response to river sediment diversions included in the Coastal Master Plan would further inform management decisions.

*Refine approach for applying research to decisions:* applied research projects often increase the knowledge base for making decisions, but it can be difficult to measurably link research projects to specific management decisions. CPRA incorporates a CPRA liaison or technical contact on funded research grants which has facilitated communication and translational research. A number of funding organizations have included advisory committees and/or science co-production as a component of research proposals that include agency staff, which helps provide conduits to connect researchers to decision makers. The approach of translating research into practice needs to be further refined to maximize application of research efforts.

*Importance of publication and value of peer-review*: CPRA publishes a large number of government reports and collects extensive monitoring datasets that provide information to managers and stakeholders. Increasing the publication of corresponding peer-reviewed literature based on analysis and synthesis of restoration and protection results would increase the robustness of applied science and processes that inform decision making on large-scale and costly projects that impact multiple stakeholders.

## 6. External Review and Evaluation

An additional process CPRA uses to incorporate new and existing knowledge into management decisions is through external evaluation. Both existing independent peer review processes and advisory review processes for specific purposes are used. Evaluations provide a mechanism for incorporating knowledge from other systems and topic-specific expertise to inform management decisions including:

- Investments needed to improve system knowledge;
- Changes needed to formalize adaptive management;
- Adjustments in planning processes to account for changing conditions.

### 6.1. External Evaluation

Science advisory committees and boards are an important component of CPRA's restoration program that is based on the best available science. Earlier text already mentioned the Coastal Master Plan Predictive Models Technical Advisory Committee, Resiliency Technical Advisory Committee, and Science and Engineering Board that provide insight into specific elements of the Coastal Master Plan process. Outside the Coastal Master Plan process, CPRA established an Expert Panel on Diversion Planning and Implementation in 2014 to provide independent advice as plans were refined for implementing sediment diversion projects along the Mississippi and Atchafalaya rivers that support coastal restoration. The Expert Panel was charged to identify critical scientific and technical uncertainties, suggest specific research to reduce uncertainty, and review and comment on technical reports, model outputs, and other aspects of project development. Meetings of the panel were structured to ensure key input was received from a variety of local experts, stakeholders, and citizens. Panel reports, wherein the Expert Panel on Diversions and Implementation provided recommendations to CPRA, which included the need for socio-economic analyses, were presented at meetings of the CPRA Board.

An external Louisiana Coastal Neotectonics and Subsidence Expert Panel was convened in 2019 under CPRA's Lowermost Mississippi River Management program to improve understanding the potential impacts of neotectonics on management of coastal resources, and better quantify variable subsidence rates across the Mississippi River Delta Plain. Presentations by regional experts were presented to the expert panel, which was charged with weighing the presented data, interpretations and conclusions [94].

An Adaptive Management Expert Panel reviewed CPRA's Adaptive Management program, and provided insights and lessons learned from the Everglades, Chesapeake Bay, Platte River, and the Columbia Estuary programs that have used an adaptive management

approach. Recommendations from the panel included lessons learned from adaptive management experience in these other ecosystems, how adaptive management was initiated and implemented in these locations, and some indication of how these lessons learned could be applied within Louisiana. The panel members shared their thoughts on some of the likely challenges in implementing adaptive management within coastal Louisiana, including large spatial scale, complexity of ecosystems and stakeholders, regulation of river flow, and highly altered landscape. For CPRA Adaptive Management, the panel suggested developing a short (<= 25 pages), clear, simple, and cost-effective adaptive management plan, with easily accessible documentation. The panel also recommended that key CPRA personnel be designated to monitor and coordinate adaptive management efforts within the agency, to train staff, and to integrate and operationalize adaptive management processes within work flows. A large report was produced collating information from both CPRA and the Deepwater Horizon Natural Resource Damage Assessment program's Louisiana Trustee Implementation Group (Appendix C), with key findings [95] including recommending coordination, a lessons learned database, ecosystem reporting, an operationalized electronic handbook, model repository, and standard operating procedures for stakeholder engagement.

*6.2. Lessons Learned from External Review and Evaluation*

*Value and benefit to reviews*: CPRA has been responsive to both internal and external evaluations, and is increasing efforts to leverage existing peer review processes, recognizing the value and benefits of both aspects of review.

*External reviews have limitations*: there can also be challenges in receiving relevant external advisory information from committees and other entities, given the complexities of the deltaic system. Moreover, external reviewers have at times not appreciated the extensive processes undertaken during project delivery, and the complex interactions between various interests and stakeholder groups.

*Exiting architecture is not data limited*: the coastal Louisiana system has extensive datasets and natural system processes knowledge. There is a need to more fully and meaningfully integrate data and vast amounts of research/science and learn from other programs.

*Translating research from knowledge to practice*: once research is complete, the process for translating research from knowledge to practice and incorporating new knowledge into project decisions is not well defined. This point emphasizes the value and role of outside scientific advice to the State's coastal program and, as addressed in Section 4, the need for a technically-robust agency staff. External scientific research is of immense value to the State as it plans and implements its risk reduction and restoration project portfolio. However, every scientific study has its caveats that may constrain consideration of its findings. The combination of outside experts advising the program and internal technical staff that can recognize those constraints and properly apply study results is critical for ensuring a scientifically-sound program and well-informed management decisions.

*Structure and governance of Adaptive Management program:* identifying the scope of an Adaptive Management program proved challenging, whether it would be CPRA-focused, or incorporate the multiple regional and Gulf funding entities (Appendix C). These are important elements to clarify early in the process. Restoration of the deltaic ecosystem encompasses preparations for a changing climate, and restoring both coastal wetlands and ecosystem services. Capturing collective institutional knowledge is important for refining program and project implementation.

*Telling the Restoration Story:* developing scientific synthesis to assess performance of management decisions and project and program implementation is important to the success of the risk reduction and restoration program. Analyzing outcomes of restoration efforts at scale and assessing effectiveness is often more complex than additive impacts of individual projects [96]. That type of systematic effort requires identifying consistent parameters and

metrics for monitoring, especially if data are used for different management purposes and/or regions.

## 7. Discussion

Louisiana's coastal natural environments are critical to the ecology of the Gulf of Mexico, the identity of the State's human communities and the nation's economy. CPRA's restoration approach integrates science into long-term coastal restoration planning to reduce and address uncertainties in determining restoration action need and the implementation of effective projects. Applied science and synthesis is incorporated at each step of the program and project development and implementation process, and includes expert input and review. Nowhere else has a restoration program of this scale been developed or implemented, and the program's success relies on using the most advanced technical information and decision support tools, and a keen understanding of the complex interactions and trade-offs that are inherent in a program of this magnitude.

Restoration in coastal Louisiana is a complex endeavor requiring the use of natural processes in a system experiencing the results of many anthropomorphic changes, including dams, levees, and changes to nutrient and sediment dynamics in the Mississippi and Atchafalaya Rivers which are being impacted upstream. The Coastal Master Plan process integrates principles such as sediment limitation, natural processes, and climate change uncertainties in a scenario approach underpinning the model projection. Riverine sediment resources can supply a portion of the needed sediment for restoration projects. However, the current riverine sediment load is insufficient for restoring the Louisiana coast to its former extent, given reduced sediment supply and changing environmental conditions such as rising sea levels [97]. Indeed, Blum and Roberts [97] concluded that without sediment addition, "significant drowning of the delta is inevitable." Both the Future Without Action and Future With Action models run substantial coastal wetland loss over the 50-year period of analysis. When compared against each other, though, the Future With Action projections estimate that potential wetland loss can be offset by as much as 3000 square kilometers, in the case of the most severe environmental scenario [22].

Louisiana's large-scale ecosystem restoration program is guided by a science-based Coastal Master Plan, and needs and opportunities for science to address technical uncertainties occur throughout the planning and project implementation process. This paper reviews how CPRA's processes incorporate science to resolve scientific and technical information needs and uncertainties at project and program scales to inform management decisions. A tremendous amount of knowledge is generated through CPRA's restoration efforts that increases understanding of the coast it is charged with protecting and restoring. Institutional knowledge gained by Louisiana from the planning and implementation of various restoration plans over the last few decades is also directly relevant to the risk reduction and restoration of resources in other states around the Gulf and to other coastal regions around the globe.

**Author Contributions:** A.M.F. conceptualization of the paper. All authors developed and implemented methods to incorporate science in planning processes. A.M.F., J.W.P., R.C.R., E.D.W. and L.A.S. undertook the formal analysis and synthesis. Manuscript writing—original draft preparation, A.M.F.; writing—review and editing, A.M.F., J.W.P., E.D.W., S.L., D.C.L., R.C.R. and L.A.S. All authors have read and agreed to the published version of the manuscript.

**Funding:** This research received no external funding.

**Institutional Review Board Statement:** Not applicable.

**Informed Consent Statement:** Not applicable.

**Data Availability Statement:** Reports and data reviewed for this article can be found here: https://cims.coastal.louisiana.gov/ (accessed on 30 April 2021).

**Acknowledgments:** We would like to acknowledge and thank John Troutman, Darin Lee, Paulina Kolic, Bill Boshart, Tommy McGinnis, Erin Plitsch, Melissa Hymel, Syed Khalil, Ed Haywood, and

other CPRA staff who have worked on technical planning processes. Additional members of the 2017 Coastal Master Plan Model Decision Team [22] provided conceptualization and implementation guidance for the 2017 Coastal Master Plan. We thank three reviewers for their comments that greatly improved the manuscript.

**Conflicts of Interest:** The authors declare no conflict of interest.

## Appendix A

*Funding*

The Coastal Master Plan is implemented yearly as described through annual plans that establish and constrain CPRA's funding priorities. CPRA projects are funded through various funding sources including the State Mineral Revenue, the Gulf of Mexico Energy Security Act, the Coastal Wetlands Planning, Protection and Restoration Act, and State Surplus Funds [98]. In 2010, the BP Deepwater Horizon Oil Spill, a tragedy that killed 11 people, was the largest marine oil spill in history [99], and impacted the Gulf of Mexico region. Coastal Louisiana was severely affected by the 4.9 million barrels of oil that were discharged into the Gulf of Mexico [100]. The total Deepwater Horizon oil spill settlement amount to Louisiana is a minimum of $7.8 billion over 15 years for ecosystem and economic restoration. Most of CPRA's funding sources have specific guidelines of how funds can be used. Oil spill settlement funds have increased CPRA's revenues, with $1.08 in total projected expenditures for Fiscal Year 2021 [98].

## Appendix B

*Building on Previous Restoration Planning Efforts in Louisiana*

With Gagliano and van Beek's research findings on coastal Louisiana land loss in the 1970s, attention started to be focused on the problem and in restoration of the coast [66,101–103]. In 1981, Louisiana Act 41 established the State's Coastal Protection Trust Fund for coastal restoration. The Caernarvon Diversion in Louisiana (Figure 4) was authorized by the Water Resources Development Act, and built in 1988–1991 in partnership with the U.S. Army Corps of Engineers, with the State's Coastal Trust Fund providing the state's share of funding; operations began in 1991. The Louisiana Legislature passed Act 6 of the Second Extraordinary Session in 1989 and established Louisiana's Wetlands Conservation and Restoration Fund, which used a portion of Louisiana's annual oil and gas severance taxes [104]. National legislation was enacted in 1990 to identify, prepare, and fund construction of coastal wetlands. The Coastal Wetlands Planning, Protection, and Restoration Act, also known as the "Breaux Act" after Louisiana Senator Breaux who championed the legislation, was the first major federal program to fund coastal wetland restoration projects (with state cost share). A restoration plan titled "Louisiana Coastal Wetlands Restoration Plan" was completed in 1993, and a multi-agency task force has undertaken studies and construction of restoration projects [105].

A recognition of the scale of the land loss problem and the necessity of large-scale efforts to address it [57] prompted collective strategic planning including Federal, State, and local governments resulting in Coast 2050: Toward a Sustainable Coastal Louisiana (Coast 2050) [84]. The Coast 2050 plan identified causes and consequences of land loss, regional ecosystem management strategies, and institutional issues. The cost of implementation of the strategies identified was approximated at $14 billion [84].

In 2004, the LCA Comprehensive Ecosystem Restoration Study, which built upon strategies identified in the Coast 2050 plan, was completed by the Corps of Engineers and the State of Louisiana. In addition to a multi-agency Project Delivery Team, over 120 scientists, engineers, and planners participated in modeling, review, and advisory capacities [103]. The CLEAR system was used to evaluate restoration alternatives, based on projections of hydrology, morphology, vegetation, and habitat suitability [71,106]. Congress, through the Water Resources Development Act (WRDA) of 2007, authorized more than $2 billion to restore wetlands in Louisiana [107].

**Appendix C**

*Appendix C.1. Gulf of Mexico Alliance*

Louisiana is a member of the Gulf of Mexico Alliance (GOMA), a regional ocean partnership intended to address technical issues germane to all five Gulf of Mexico states. Led by those five states, GOMA has a large network including federal agencies, academic institutions, businesses, and non-profits, working collaboratively to enhance the environmental and economic health of the region. The current structure of GOMA has multi-agency teams that are addressing technical aspects associated with coastal community resilience, habitat resources, water resources, and wildlife and fisheries, along with data and monitoring and education and engagement topics. GOMA has provided the State with access to resources to address some of our recognized technical uncertainties (where they overlap with those of the other four Gulf states), and conversely allows for both the communication of State efforts to neighboring states and provides an opportunity for State technical leadership on multi-state efforts.

*Appendix C.2. Gulf Coastal Ecosystem Restoration Council Monitoring and Assessment Working Group*

In July 2012, the Resources and Ecosystems Sustainability, Tourist Opportunities, and Revived Economies of the Gulf Coast States Act (RESTORE Act) established the Gulf Coast Ecosystem Restoration Council (Council), which has oversight of 60% of expenditures from the Deepwater Horizon Spill Gulf Coast Restoration Trust Fund. The RESTORE Council Monitoring and Assessment Working Group (CMAWG) serves as the leadership body responsible for coordinating Council monitoring activities, including the recommendation of monitoring and assessment standards that will be used for Council projects and programs. The CMAWG consists of primary and secondary representatives from the 11 RESTORE Council members (States of Florida, Alabama, Mississippi, Louisiana, and Texas; U.S. Departments of Agriculture, Interior, Commerce and Homeland Security; U.S. Army Corps of Engineers; and the Environmental Protection Agency). CPRA is the Louisiana state agency with representatives on the CMAWG.

*Appendix C.3. Natural Resources Damage Assessment*

The Louisiana Trustee Implementation Group (LA TIG) represents a joint effort between the State of Louisiana (CPRA, Louisiana Oil Spill Coordinator's Office, Louisiana Department of Environmental Quality, Louisiana Department of Wildlife and Fisheries, Louisiana Department of Natural Resources) and the federal trustees (Department of Commerce: National Oceanic and Atmospheric Administration, Department of the Interior, U.S. Environmental Protection Agency, and the U.S. Department of Agriculture) to evaluate the impacts of the BP Deepwater Horizon Oil Spill in Louisiana, and to plan and carry out restoration efforts. The LA-TIG developed restoration plans for Louisiana resources impacted by the BP oil spill [108]. A portion of Natural Resources Damage Assessment (NRDA) funds to implement restoration in Louisiana are for Monitoring and Adaptive Management (MAM), and the LA TIG initiated a process to clarify decision making processes to guide future spending of these MAM funds. CPRA is leading the process (with other LA TIG members) of developing objectives for each NRDA restoration type, including wetlands coastal and nearshore habitat and other restoration types to provide a decision-support tool for funding MAM activities.

Louisiana (including technical representatives from CPRA) also participates, along with other Gulf states and federal entities, in a Cross-Trustee Implementation Group (Cross-TIG) Monitoring and Adaptive Management (MAM) work group. This group was established to meet monitoring and Adaptive Management responsibilities laid out in the Programmatic Damage Assessment and Restoration Plan and the Final Programmatic Environmental Impact Statement and the Trustee Council Standard Operating Procedures.

*Appendix C.4. Mississippi River/Gulf of Mexico Hypoxia Task Force*

The role of the Mississippi River/Gulf of Mexico Hypoxia Task Force (HTF) is to provide a working collaboration among states, federal agencies, and tribes to reduce nutrient pollution in the Mississippi/Atchafalaya River Basins and the extent of the hypoxic zone in the Gulf of Mexico. The goal of the HTF is to reduce the hypoxic zone to less than 5000 $km^2$ by 2035 with an interim target to reduce nitrogen and phosphorus loading 20% by 2025 (relative to the 1980–1996 baseline average loading to the Gulf). The 2020 Gulf of Mexico hypoxic zone measured 5489 $km^2$, a relatively smaller size due in large part to mixing caused by Hurricane Hanna [109]. The 5-year average size is 14,007 $km^2$. The HTF is a partnership of 12 states along the Mississippi River (including Louisiana), five federal agencies, and a representative for Tribal Nations.

*Appendix C.5. Synergies with Regional-Scale Efforts*

CPRA research priorities were shared with the National Academies of Science Gulf Research Program and the Gulf of Mexico Alliance, and incorporated into their respective strategic plans [110,111], broadening the potential funding options for addressing those priorities. CPRA technical staff serve on advisory committees for numerous Louisiana research projects funded by external entities such as Louisiana Sea Grant, the National Oceanic and Atmospheric Administration's RESTORE Science Program, the National Academies of Sciences, Engineering, and Medicine Gulf Research Program, and the National Science Foundation, as well as the program advisory boards themselves.

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
