# Peer review of "A Review of How Uncertainties in Management Decisions Are Addressed in Coastal Louisiana Restoration"

_water, doi:10.3390/w13111528_

Round 1

Reviewer 1 Report

Although this manuscript is significant in that it synthesizes long-term project outcomes, it is considered inappropriate to be published as a article. At least from my point of view, a article should be able to lead discussions by limiting the scope of the research topic and supporting a clear scientific basis for the topic. However, this manuscript does not reveal the academic value of the article mentioned above.

Author Response

Comment 1: Although this manuscript is significant in that it synthesizes long-term project outcomes, it is considered inappropriate to be published as a article. At least from my point of view, a article should be able to lead discussions by limiting the scope of the research topic and supporting a clear scientific basis for the topic. However, this manuscript does not reveal the academic value of the article mentioned above.

Response 1: Thank you for the review and comment. We agree, and revised the article into a review paper to better reflect its intent. Also, we majorly revised the paper to focus the review on how uncertainties in management decisions are addressed in coastal Louisiana restoration. The application of science to inform coastal restoration decision making and support the strategic implementation of restoration programs and projects in Louisiana is pioneering. The processes developed are described and integrated in lessons learned sections to chronicle and evaluate planning methods in Louisiana, and to provide guidance on adaptation planning in other vulnerable coastal regions.    

Reviewer 2 Report

The paper is well written and gives an interesting review of the geographical context of the Mississippi Delta area and of the related management and ecosystem restoration projects.

The description of the processes that Louisiana’s Coastal Protection and Restoration Authority (CPRA) uses make science-based decisions is exaustive but a bit long and could be difficult to follow for someone that does not know the local plannings and projects contest. For this reason I suggest to insert two figures: 1) a map showing the most important toponyms (river diversions, lakes, etc...) used in the text; 2) a figure/table with a schematic presentation of the projects and plans discussed in the paper, to help the reader to understand the hierarchy and the sequence of the steps you are following.

All the figures need to be cited in the text (references).

Other minor comments are in the pdf.

Author Response

Comment 1: The paper is well written and gives an interesting review of the geographical context of the Mississippi Delta area and of the related management and ecosystem restoration projects.

Response 1: Thank you for this comment.

Comment 2: The description of the processes that Louisiana’s Coastal Protection and Restoration Authority (CPRA) uses make science-based decisions is exaustive but a bit long and could be difficult to follow for someone that does not know the local plannings and projects contest. For this reason I suggest to insert two figures: 1) a map showing the most important toponyms (river diversions, lakes, etc...) used in the text; 2) a figure/table with a schematic presentation of the projects and plans discussed in the paper, to help the reader to understand the hierarchy and the sequence of the steps you are following.

Response 2: Two new figures were added. (Figure 4 and Figure 5). Figure 4 includes the toponyms used in the text, and also includes monitoring data collected/implemented to data for additional context. The second new figure (Figure 5) replaces previous Figure 6, and incorporates the information from previous Figure 6 into a schematic of the planning processes CPRA uses, and how science is incorporated to reduce uncertainties. In addition, the paper was revised into a review paper that focuses on how uncertainties in management decisions are addressed in coastal Louisiana restoration. Descriptions of the processes CPRA uses were clarified and shortened, and some detailed information moved to appendices to improve readability. Lessons learned sections were incorporated to synthesize and evaluate each of the high-level planning processes.   

Comment 3: All the figures need to be cited in the text (references).

Response 3: All figures are referenced in the Word version of the manuscript. The PDF version of the manuscript the reviewer reviewed has “(Error! Reference source not found.)” instead of the figure references. This may be related to figure cross referencing used in the Word text, and therefore the cross referencing functionality was removed, and only the “Figure X” text is incorporated. 

Comment 4: Other minor comments are in the pdf.

Response 4: Comments from the PDF are copied into the attached Word file, and responses provided. Please see the attachment. 

Reviewer 3 Report

See attached file.

Author Response

Reviewer 3

General Comment: The manuscript of the article “Coastal Louisiana Restoration: Addressing Uncertainties in Management Decisions” by Freeman et al., presents the history, background and results of “Louisiana’s Comprehensive Master Plan for a Sustainable Coast”.

This work is an exhaustive presentation of the master plan in terms of: (a) the presentation and history of the study area, (b) the presentation and history of the management authorities’ approaches to the various problems faced in the wider area, (c) the presentation of the structure and contents of the plan. It is more than evident that a lot of effort has been put behind this work.

However, in its present form, there are a series of critical issues to be resolved prior to it being considered for publication as a research article in Water. These issues are highlighted in the remarks to be found in the following (I do not elaborate too much on specifics and examples from the text, as this would have made the review quite extensive). If the authors feel that they can successfully address them, I would gladly contribute to the review process of a revised version of this work.

Response General Comment: Thank you for the review and comments.  We agree, and revised the article into a review paper to better reflect its intent. Also, we majorly revised the paper to focus the review on how uncertainties in management decisions are addressed in coastal Louisiana restoration. The application of science to inform coastal restoration decision making and support the strategic implementation of restoration programs and projects in Louisiana is pioneering. The processes developed are described and integrated in lessons learned sections to chronicle and evaluate planning methods in Louisiana, and to provide guidance on adaptation planning in other vulnerable coastal regions.    

We included the additional remarks and our responses in the attached Word document. Please see the attachment. 

Round 2

Reviewer 1 Report

I reviewed that the type of manuscript was changed to a review article, and that the authors have modified a significant amount of the manuscript. This manuscript is significant in implying a long and extensive Coastal Louisiana Restoration Project. However, I would like to recommend that this manuscript be more in the form of a scientific paper. Through the composition of independent research method chapters, authors should clearly present the spatial, temporal, and content scope of this project covered in the paper, and the research materials and methods for review in each field of the project should be specified. This work will guide the reader in understanding the structure and content of this paper.